

# Phylogenetic and morphologic evidence confirm the presence of a new montane cloud forest associated bird species in Mexico, the Mountain Elaenia (*Elaenia frantzii*; Aves: Passeriformes: Tyrannidae)

Zachary R. Hanna[1,2,3], Marco F. Ortiz-Ramírez[4,5],
César A. Ríos-Muñoz[4,6], Héctor Cayetano-Rosas[4,5],
Rauri C. K. Bowie[1,2] and Adolfo G. Navarro-Sigüenza[4]

[1] Museum of Vertebrate Zoology, University of California, Berkeley, Berkeley, California, United States of America
[2] Department of Integrative Biology, University of California, Berkeley, Berkeley, California, United States of America
[3] Ornithology & Mammalogy, California Academy of Sciences, San Francisco, California, United States of America
[4] Museo de Zoología, Facultad de Ciencias, Universidad Nacional Autónoma de México, México, Distrito Federal, México
[5] Posgrado en Ciencias Biológicas, Universidad Nacional Autónoma de México, México, Distrito Federal, México
[6] Unidad de Investigación en Medicina Experimental, Facultad de Medicina, Universidad Nacional Autónoma de México, México, Distrito Federal, México

Corresponding author
Zachary R. Hanna,
zachanna@berkeley.edu

## ABSTRACT

Here we provide evidence to support an extension of the recognized distributional range of the Mountain Elaenia (*Elaenia frantzii*) to include southern Mexico. We collected two specimens in breeding condition in northwestern Sierra Norte de Chiapas, Mexico. Morphologic and genetic evidence support their identity as *Elaenia frantzii*. We compared environmental parameters of records across the entire geographic range of the species to those at the northern Chiapas survey site and found no climatic differences among localities.

## INTRODUCTION

The Mountain Elaenia (*Elaenia frantzii*) is a small New World flycatcher that breeds in the high elevation forests of Central America and northern South America and is known to move seasonally to lower elevations (*Stiles & Skutch, 1989*). Its distribution has previously been reported to include isolated populations extending from Colombia and Venezuela as far north as north-central Guatemala (*Land, 1970*; *Howell & Webb, 1995*; *Eisermann & Avendaño, 2007*), but it has not been documented in Mexico (*Peterson & Chalif, 1973*; *Hosner, 2004*). Its habitat consists of a variety of open humid to semi-humid forest types, including forest edges, secondary growth, and farmland, in an altitudinal range of

750–3600 m. A continuum of fragmented cloud forest extends from the east of the Isthmus of Tehuantepec to the Nicaraguan depression (*Ramírez-Barahona & Eguiarte, 2014*), providing potential habitat for this bird in southern Mexico.

We analyzed the morphological and molecular characteristics of two specimens collected in Chiapas, Mexico, during August 2013 and compared them with the associated data from museum specimens of other *E. frantzii*. We additionally investigated environmental parameters from the previously known distribution of *E. frantzii* and compared them with those present in southern Mexico to assess the likelihood that suitable conditions exist in southern Mexico to support a resident population of this species.

## METHODS

### Taxon morphological description

*Elaenia frantzii* is a small tyrant flycatcher with a rounded head lacking a coronal patch. It has dark brown irides and narrow, pale-lemon eye-rings. Its bill is black with a pale, flesh-colored base of the lower mandible and a dusky culmen and tip. The head and upperparts are brownish olive, the throat and chest are grayish olive, and the belly and undertail coverts are pale-lemon. The wing is dusky with conspicuous yellowish wing-bars. The edges of the remiges are yellowish and the tertial edgings are white. The legs are blackish. Males and females have similar external morphology (*Howell & Webb, 1995*; *Hosner, 2004*).

*Elaenia frantzii* has a patchy distribution throughout the mountain systems of Central and South America. It inhabits open humid to semi-humid forest, forest edges, secondary growth, and farmland areas where scattered bushes and trees are present. Based on geographic distribution and subtle differences in plumage color and body size, four *E. frantzii* subspecies are recognized (*Hosner, 2004*).

The nominate subspecies, *E. f. frantzii Lawrence, 1867*, is found in Nicaragua, Costa Rica, and Panama (*Hosner, 2004*). *Elaenia frantzii pudica Sclater, 1871* is overall smaller, darker above, and paler below than the nominate subspecies and inhabits the Andes mountains of northern Colombia and Venezuela (*Hosner, 2004*). *Elaenia frantzii browni Bangs, 1898* is restricted to the Sierra Nevada de Santa Marta of northern Colombia and the Sierra de Perijá of northwestern Venezuela and is yet smaller and with paler upperparts than *E. frantzii pudica* (*Hosner, 2004*). *Elaenia frantzii ultima Griscom, 1935* is a subspecies darker than the nominate inhabiting Guatemala, Honduras and El Salvador and is the most proximately distributed subspecies to southern Mexico (*Hosner, 2004*). See Fig. 1 for a map of the distributions of the subspecies.

*Elaenia frantzii* can be distinguished from the Yellow-bellied Elaenia (*E. flavogaster*), a Mexican resident species, by the lack of a white coronal patch, more greenish coloration of the upperparts and less extended yellow in the underparts. Although the coloration is similar to some members of the genus *Empidonax*, *E. frantzii* differs from them in its narrow eye-ring and the relative proportions of the body, head and bill.

### Study area

As members of a joint Museum of Vertebrate Zoology (MVZ), University of California, Berkeley and Museo de Zoología "Alfonso L. Herrera" (MZFC), Facultad de Ciencias,

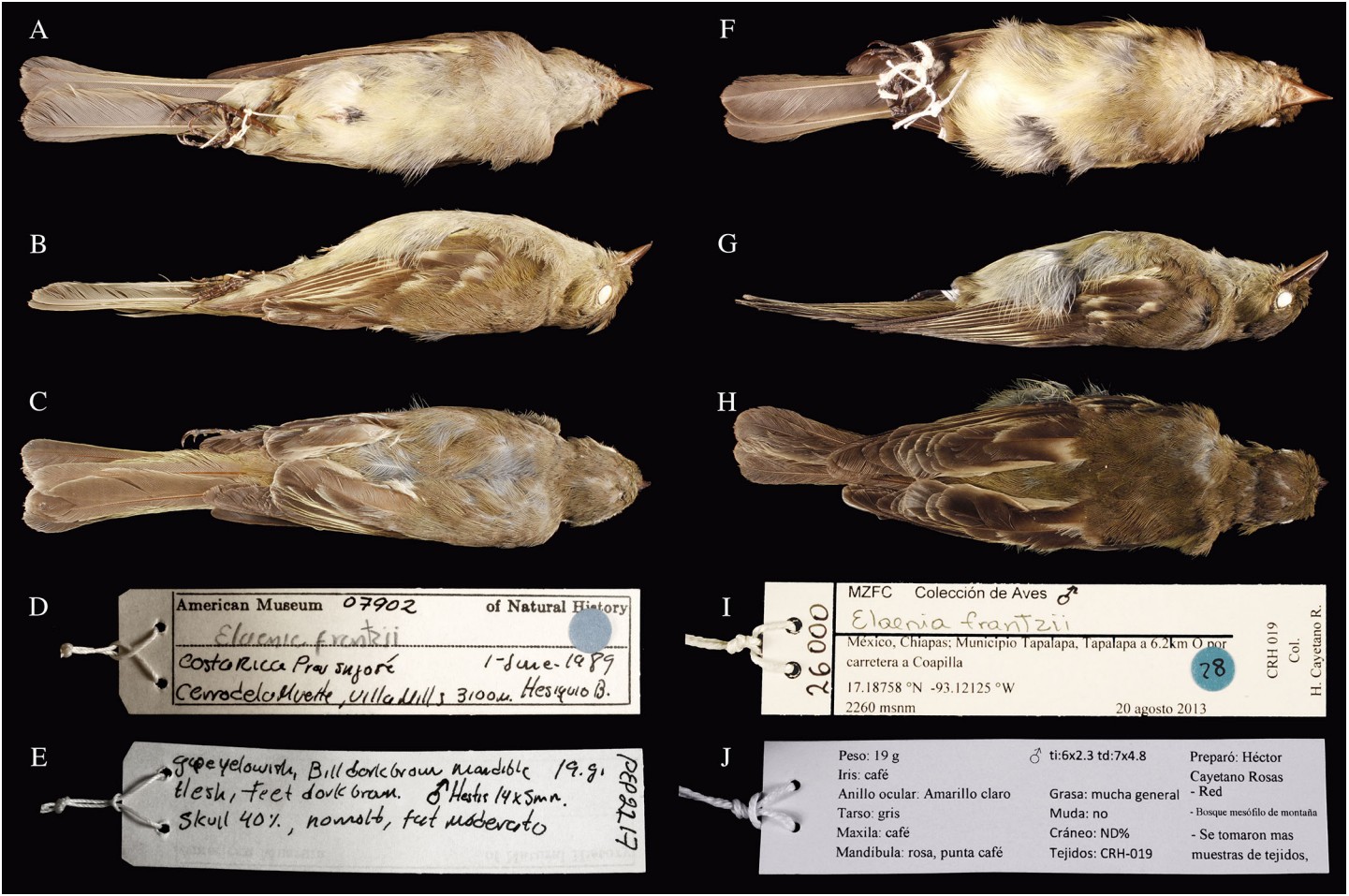

**Figure 1 Ventral, lateral and dorsal views of two *Elaenia frantzii*.** Panels (A–E) a specimen of *E. f. frantzii* collected in Costa Rica (AMNH: Birds:07902: Table 1). Panels (F–J) a specimen of *E. frantzii* collected in Chiapas, México (MZFC:26000). Note the pronounced darker coloration of the Chiapas specimen, especially in the dorsal and lateral views.

Universidad Nacional Autónoma de México team formed to survey the bird, mammal, and herpetological diversity at montane sites in southern Mexico, we traveled to a tract of cloud forest 6.2 km W (by road) from the center of Tapalapa on the road to Coapilla, Municipio Tapalapa, Chiapas, Mexico in August 2013. We sampled the avifauna of the site using mist-nets from 20–24 August 2013 under field permit FAUT-0169 granted by the Mexican government agency Secretaría de Medio Ambiente y Recursos Naturales and with animal ethical approval R317 granted by the UC Berkeley Institutional Animal Care and Use Committee. The mist-nets were 6–12 m long, 2.5 m high, and were made of 16 mm × 16 mm mesh. We preserved a portion of the captured individuals as voucher specimens and deposited them in ornithological research collections at the Museum of Vertebrate Zoology (MVZ) and Museo de Zoología "Alfonso L. Herrera" (MZFC). We collected tissue samples and assessed the breeding condition of each voucher specimen by measuring gonads and recording any evidence of a brood patch or cloacal protuberance. We also recorded data on molt, fat, skull ossification, and parasite load. After returning

from the field, we confirmed the species status of the collected specimens by comparing their preserved external morphologies with individuals deposited in the MZFC.

## Bibliographic search

We conducted all searches described below on 20 December 2014. We performed a literature search for published observations of *E. frantzii* in Mexico. We additionally searched for records of *E. frantzii* in VertNet (http://www.vertnet.org), a web-based biodiversity data aggregator. We also searched eBird (*Sullivan et al., 2009*; http://www.ebird.org), an online checklist program, which archives lists of bird sightings submitted by professional and amateur birdwatchers, for *E. frantzii* sightings most proximate to Mexico. As an assessment of whether a lack of documentation of this species in Chiapas, Mexico might be a result of low sampling effort or due to this species having recently expanded its range into Chiapas, we performed a search in VertNet (http://www.vertnet.org) on 20 December 2014 with Record Type = "Any type" and using the following Darwin Core terms: Class = "Aves," Country = "Mexico," and StateProvince = "Chiapas."

## Laboratory protocol

We gathered genetic evidence to support our morphology-based identifications by sequencing 1041 bp of the mitochondrial locus Nicotinamide Adenine Dinucleotide Dehydrogenase Subunit 2 (*ND2*). We first extracted genomic DNA from ~25 mg of liver tissue using a DNeasy Blood & Tissue Kit (Qiagen, Hilden, Germany). We then amplified the mitochondrial locus *ND2* using the primers L5204 and H6312 (*Cicero & Johnson, 2001*). Polymerase chain reaction (PCR) conditions included an initial denaturation at 94 °C for 3 min; then 30 cycles at 94 °C for 30 s, 54 °C for 30 s, and 72 °C for 1 min 30 s; and a final extension at 72 °C for 10 min. We sequenced PCR products in both forward and reverse directions using BigDye terminator chemistry (Applied Biosystems, Foster City, CA, USA) on an ABI 3730 automated sequencer (Applied Biosystems, Foster City, CA, USA). We edited and constructed consensus sequences using Geneious version 7.1.7 (*Kearse et al., 2012*; http://www.geneious.com).

## Maximum likelihood tree

We retrieved complete *ND2* gene sequences for 20 additional individuals representing 10 tyrannid taxa from GenBank (accession numbers appended to the end of taxon names in Fig. 2). We aligned sequences using MUSCLE (*Edgar, 2004*) and constructed a maximum likelihood tree using RAxML version 8.1.20 (*Stamatakis, 2014*). We used the GTRGAMMA model, searched for the tree with the highest likelihood, and estimated node support using 1000 bootstrap pseudoreplicates.

## Environmental characterization and species distribution model

We downloaded all available georeferenced *E. frantzii* occurrence records from the Global Biodiversity Information Facility (http://www.gbif.org). We constrained our examination of ecological parameters to the area accessible to the species (*Peterson et al., 2011*). We considered this to be the area from the Isthmus of Tehuantepec, Mexico, to northwestern South America including the Andes Mountains from Ecuador to Northern Venezuela,

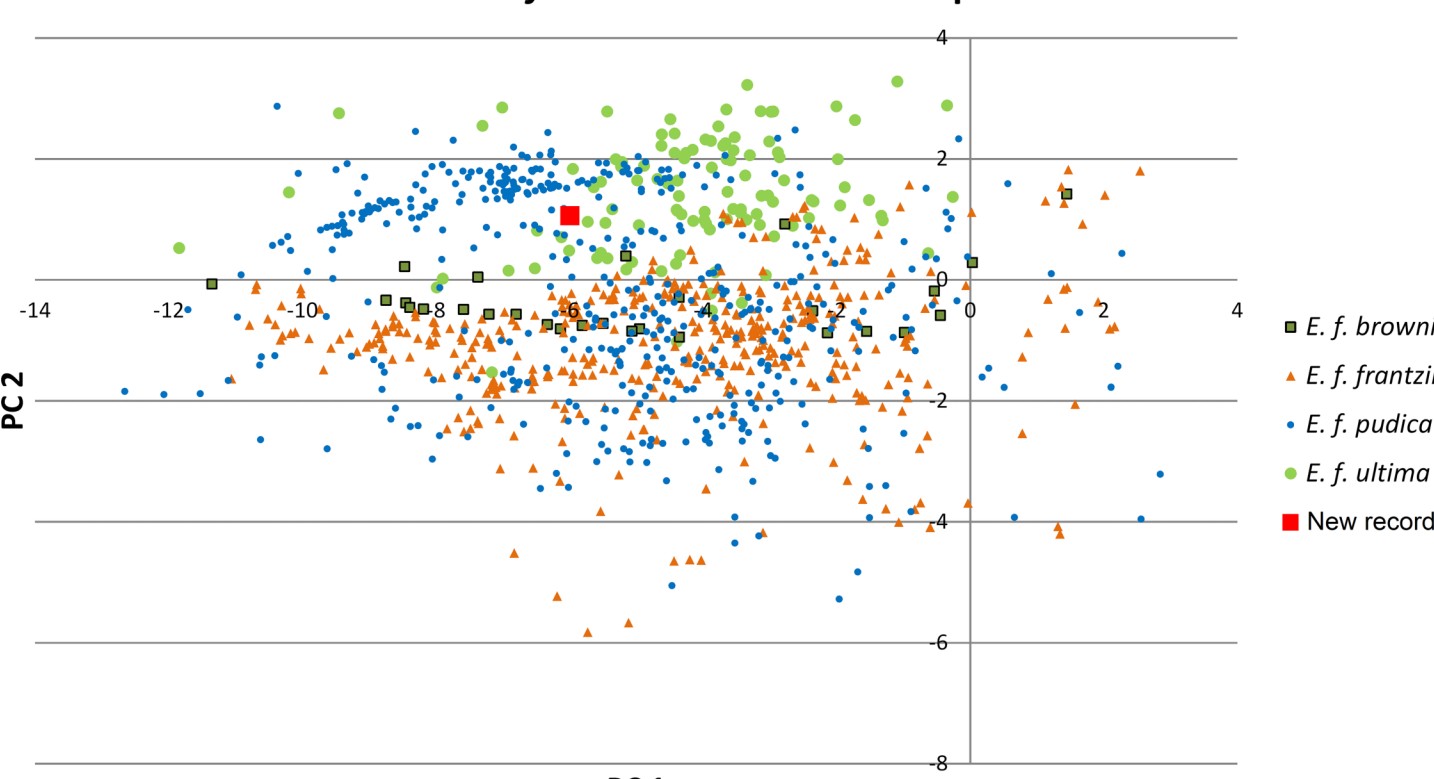

## *Elaenia frantzii* environmental space

**Figure 2 Principal components plot of environmental parameters at sites of *Elaenia frantzii* records.** Green squares: *E. frantzii browni* of extreme northern South America; orange triangles: *E. f. frantzii* of Costa Rica and Panama; blue circles: *E. frantzii pudica* of northern South America; green circles: *E. f. ultima* of northern Central America; red square: new *E. frantzii* record from Chiapas, México.

which is the expanse that includes all vouchered records available for the species. We used a set of 19 bioclimatic variables obtained from the WorldClim project (*Hijmans et al., 2005*; http://www.worldclim.org) and five topographic variables; obtained from the Hydro1k project (https://lta.cr.usgs.gov/HYDRO1K), which summed to 24 variables to characterize the environment of the potential distribution of *E. frantzii* (Table 1).

We summarized the ecological variation across the distribution of *E. frantzii* by performing a principal component analysis (PCA) on the values of the 24 ecological variables from the sites of georeferenced *E. frantzii* occurrence records using the package ENMGadgets (*Barve & Barve, 2014*) in R version 3.2.2 (*R Core Team, 2015*). This reduced the multidimensionality of the 24 ecological variables. We then associated all of the records of *E. frantzii* with the PCA output, obtained PCA values for each locality, and plotted PC1 and PC2 of each *E. frantzii* record.

We employed Maxent 3.3.3k (*Phillips, Anderson & Schapire, 2006*) to model the distribution of *E. frantzii* and assess the suitability of the new Chiapas location as compared with the sites of other *E. frantzii* occurrences. We used the package raster (*Hijmans, 2015*) and the implementation of the Pearson correlation index in R version

**Table 1** *Elaenia frantzii* **specimen data.** We here provide further information regarding the datasets that archive the *Elaenia frantzii* specimens to which we refer throughout the manuscripit.

| Specimen | Data publisher | Date accessed | Link to dataset |
|---|---|---|---|
| AMNH:Birds:07902 | AMNH Birds, American Museum of Natural History | 20 December 2014 | http://ipt.vertnet.org:8080/ipt/resource.do?r=amnh_birds |
| BLB:Recordings:27930 | BLB Recordings, Borror Laboratory of Bioacoustics, Ohio State University | 20 December 2014 | http://hymfiles.biosci.ohio-state.edu:8080/ipt/resource.do?r=blb |
| XC256448 | R.C. Hoyer. Xeno-canto | 20 October 2015 | www.xeno-canto.org/256448 |
| XC256449 | R.C. Hoyer. Xeno-canto | 20 October 2015 | www.xeno-canto.org/256450 |
| XC256450 | R.C. Hoyer. Xeno-canto | 20 October 2015 | www.xeno-canto.org/256450 |
| CLO:ML:140002 | Macaulay Library Audio and Video Collection, Cornell Lab of Ornithology | 20 December 2015 | http://ipt.vertnet.org:8080/ipt/resource.do?r=cuml_sound_film |
| USNM:Birds:69831 | USNM Birds, National Museum of National History, Smithsonian Institution | 20 December 2015 | http://ipt.vertnet.org:8080/iptstrays/resource.do?r=nmnh_birds |
| USNM:Birds:80404 | USNM Birds, National Museum of National History, Smithsonian Institution | 20 December 2015 | http://ipt.vertnet.org:8080/iptstrays/resource.do?r=nmnh_birds |

3.2.2 (*R Core Team, 2015*) to select a subset of 13 of the 24 bioclimatic and topographic variables with correlation values below 0.85 (Table 1) to avoid over parameterization of the model (*Elith et al., 2006*; *Elith & Leathwick, 2007*). We used the 13 environmental variable subset and all of the georeferenced *E. frantzii* occurrence records in Maxent 3.3.3k (*Phillips, Anderson & Schapire, 2006*) to generate ten model replicates (using the subsample replicated run type) using 30% of the sample records to test the variability produced by a random selection of the training data (random seed). We obtained suitability values from the Maxent model and mapped them along with all of the *E. frantzii* occurrence records using ArcMap 10.0 (ESRI).

## RESULTS

### Bibliographic search

We found no published observations of *E. frantzii* in Mexico. The northernmost record of this species in VertNet is a 1.41 s audio recording of a call from northeast Utah, United States of America made in May 1999 (BLB:Recordings:27930: Table 1). This record is likely a misidentification or, if not, it documents a vagrant and does not indicate a breeding population as there are no other records of *E. frantzii* within 3000 km

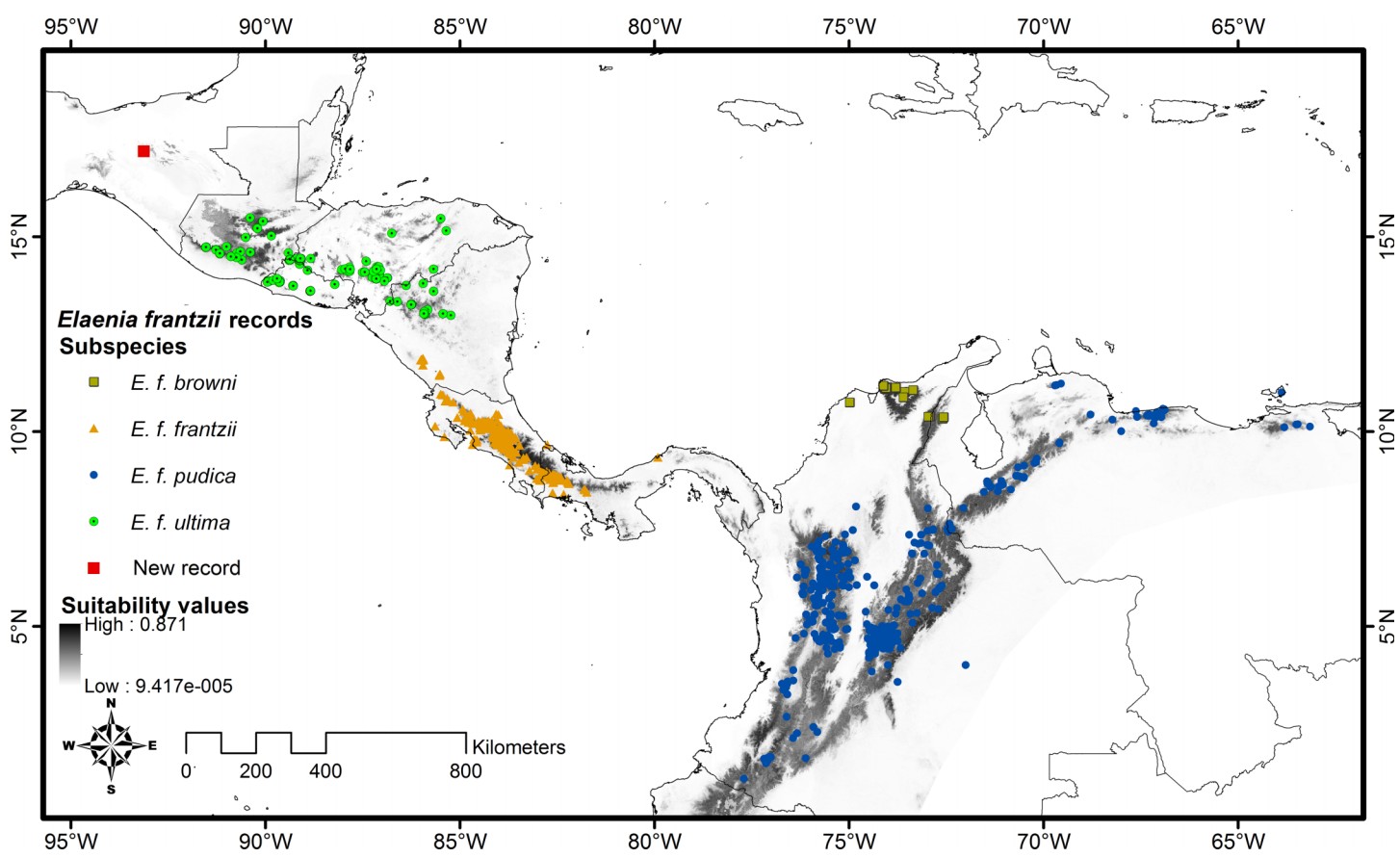

**Figure 3 Distribution map of *Elaenia frantzii* with suitability values displayed geographically.** Gray shading indicates the geography of the suitability values predicted by the Maxent model. Darker shades indicate areas with higher suitability while lighter shades denote areas with lower suitability. We have mapped *E. frantzii* records on top of the suitability predictions and color-coded them by subspecies. The red square indicates the site of the new records in Chiapas, Mexico.

of this location. The records of *E. frantzii* closest to Mexico along the land axis of Central America are in southwestern Guatemala (CLO:ML:140002, USNM:Birds:69831, USNM:Birds:80404: Table 1). The sightings closest to Mexico publicly visible in eBird (*Sullivan et al., 2009*; http://www.ebird.org) as of 20 December 2014 were made in July 2001 at 14.7173609°N and 91.5309191°W, WGS84, Santa María de Jesús, Departamento Quetzaltenango, Guatemala (*Berry, 2001*) and in February 2013 at 15.4769771°N and 90.3913987°W, WGS84, Cobán, Departamento Alta Verapaz, Guatemala (*Cahill, 2013*). Figure 3 includes these records. Our class Aves-based VertNet search indicated that there have been avian records from montane regions in Chiapas from every decade spanning the years 1880–2009, but *E. frantzii* has never been detected in this region.

## Morphological identification

We surveyed for a total of 283.39 12-m net hours and captured two individuals, which we gave a preliminary designation of *E. frantzii* based on plumage and morphology and assigned to the collector numbers CRH019 and ZRH814. We captured CRH019 on

20 August 2013. It did not display any evidence of molt and was a male in breeding condition with a right testis of $7 \times 4.8$ mm. We captured ZRH814 on 23 August 2013. ZRH814 was a female in breeding condition with an $8 \times 4$ mm ovary containing two collapsed follicles. It had extensive fat layers, a fully ossified skull, and no evidence of molt.

We deposited a round skin voucher specimen and tissue sample of CRH019 in MZFC as MZFC:26000. We archived a tissue sample of CRH019 in MVZ as MVZ:Birds:188004. We deposited a round skin voucher specimen and tissue sample of ZRH814 in MVZ as MVZ:Birds:188003. We archived a tissue sample of ZRH814 in MZFC as ZRH814.

We confirmed the identity of the Chiapas specimens through comparison with *E. frantzii* museum skins. Compared with the *E. f. frantzii* specimen (AMNH:Birds:07902: Table 1) collected in Costa Rica, the newly collected specimens are darker in the upperparts, tail, and wings. The chest and flanks are more olive-colored and darker overall. The edges of the remiges are narrower and contain less yellow. The bill is wider at the distal edge of the nostrils (4 mm in our specimen vs. 3 mm in the Costa Rica specimen) (Fig. 3). These differences are in agreement with the description of the subspecies *ultima* compared to *frantzii* (*Hosner, 2004*).

## Molecular identification

We obtained the complete *ND2* gene sequence, 1041 bp, from specimens MZFC:26000 and MVZ:Birds:188003, which shared the same *ND2* haplotype. We deposited newly generated sequences in GenBank (accession numbers KU312259–KU312260). The maximum likelihood tree grouped the Chiapas *E. frantzii ND2* sequence with sequences of *E. frantzii* from El Salvador and Panama (GenBank accession numbers EU311059 and EU311049, respectively) with 100% bootstrap support (Fig. 4).

## Environmental characterization and species distribution model

The first two principal components of the 24 bioclimatic and topographic variables included 63.24% of the cumulative ecological variance for the species. The variables which have higher eigenvectors for component 1 are mainly associated with temperature variables whereas those for component 2 are primarily associated with precipitation (Table 2). Values of the environmental variables at the new Chiapas locality fall within the variation of those variables at *E. frantzii* occurrence sites in populations from Central America and South America (Fig. 4).

We chose the median Maxent model of the ten replicates we generated in order to account for the variation produced by the random selection of sample record seeds in each replicate. The suitability value at the site of the new record of *E. frantzii* in Chiapas is 0.24. The highest suitability value in our model was 0.87. The sites of 94% of the *E. frantzii* records have suitability values greater than or equal to 0.24 (Fig. 3). Georeferencing error, sink populations, or vagrant birds may account for a portion of the *E. frantzii* locations situated at sites with low suitability (*Peterson et al., 2011*).

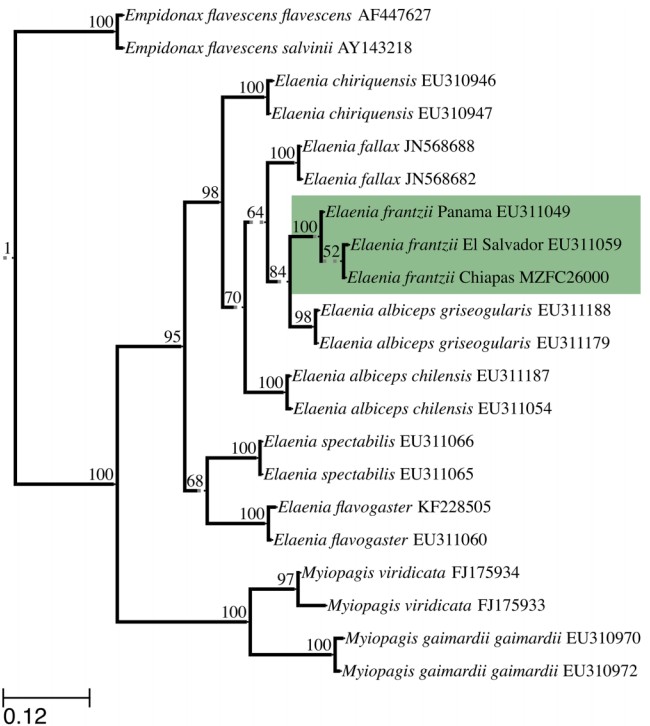

**Figure 4 Maximum likelihood topology constructed with *ND2* sequences from *E. frantzii* and other tyrant flycatchers (Tyrannidae).** We have highlighted the *E. frantzii* clade in green. We labeled branches with bootstrap support values (%) from 1000 pseudoreplicates. Grayed, broken lines denote branches that have been artificially elongated for the sake of clarity in the figure. We rooted the topology with the common ancestor of the *Empidonax* group. The scale bar indicates the branch-lengths that correspond to the maximum likelihood estimate of the number of substitutions that have occurred on average per site between two nodes in the tree.

## DISCUSSION

Morphological and molecular evidence confirm the presence of a breeding population of *E. frantzii* in Mexico. These records warrant an extension of the range of *E. frantzii* to include the state of Chiapas, Mexico. The closest locality of a publicly available, georeferenced voucher (an audio voucher in this instance) to our new records is 360 km away (by air) in Guatemala (140002, Macaulay Library Audio and Video Collection, Cornell Lab of Ornithology). Our collection site is 190 km from the Guatemalan border in the northwestern Sierra Norte de Chiapas at the northern limit of Nuclear Central America (NCA), the area between the Isthmus of Tehuantepec and the Nicaraguan depression (*Schuchert, 1935*). It is not surprising that *E. frantzii*'s range could include the extreme northern portion of NCA as its previously known range includes much of the rest of NCA, across which environmental conditions are largely similar (*Ramírez-Barahona & Eguiarte, 2014*). Our ecological niche analyses indicate that climatic conditions where *E. frantzii* is found in Central and South America are present in Chiapas, Mexico. This distributional pattern of inhabiting cloud forests from Chiapas south through NCA is exhibited by other species of montane birds, such as *Chlorospingus ophthalmicus, Catharus frantzii,* and *Atlapetes albinucha* (*Howell & Webb, 1995*), which we

**Table 2 Environmental variables and their eigenvectors from the principal component analysis (PCA).** PC1 and PC2 refer to the first two principal components of the PCA. We have highlighted the three highest loading values for each principal component in bold. Variables with a trailing asterisk.

| Original environmental variables | PC1 | PC2 |
|---|---|---|
| Annual mean temperature | **3.34E-01** | 2.98E-04 |
| Mean diurnal range | −2.25E-02 | 1.79E-01 |
| Isothermality | −1.03E-01 | −2.05E-01 |
| Temperature seasonality | 8.24E-02 | 1.73E-01 |
| Maximum temperature of warmest month* | 3.27E-01 | 5.96E-02 |
| Minimum temperature of the coldest month* | 3.19E-01 | −6.72E-02 |
| Temperature annual range* | 4.07E-02 | 2.49E-01 |
| Mean temperature of wettest quarter | 3.31E-01 | 1.41E-02 |
| Mean temperature of driest quarter | **3.32E-01** | −1.65E-02 |
| Mean temperature of warmest quarter* | **3.35E-01** | 2.15E-02 |
| Mean temperature of coldest quarter* | 3.26E-01 | −2.53E-02 |
| Annual precipitation* | 4.48E-02 | **−3.66E-01** |
| Precipitation of wettest month* | 7.53E-02 | −2.92E-01 |
| Precipitation of driest month | −2.19E-02 | **−3.50E-01** |
| Precipitation seasonality | 8.91E-02 | 2.66E-01 |
| Precipitation of wettest quarter | 7.97E-02 | −3.05E-01 |
| Precipitation of driest quarter* | −1.99E-02 | **−3.60E-01** |
| Precipitation of warmest quarter | −4.20E-02 | −3.04E-01 |
| Precipitation of coldest quarter | 7.19E-02 | −3.10E-01 |
| Altitude* | −3.26E-01 | 5.18E-03 |
| Slope* | −2.29E-01 | 6.67E-03 |
| Eastness slope* | −7.92E-03 | −5.50E-03 |
| Northness slope* | 5.48E-03 | 4.93E-03 |
| Topographic index* | 1.83E-01 | −5.61E-03 |

**Note:**
*Comprise the 13 variables with Pearson correlation index below 0.85 used in the Maxent modeling.

also recorded at this locality. After we completed our literature searchers and began to prepare this manuscript, a group of bird watchers visited several sites in the vicinity of our survey site near Tapalapa, Chiapas, México on 7–13 July 2015. The birders sighted multiple *Elaenia frantzii* (*Martinez, 2015*; *Hoyer, 2015*) and made several recordings of their calls (XC256448, XC256449, XC256450: Table 1). These detections indicate that the population that we surveyed appears to be persisting.

It is remarkable that this species was previously unknown to Chiapas, despite sampling having been conducted at montane sites in the area over the past century. It is possible that this species has been misidentified in past surveys and collections due to its resemblance to other flycatcher species in the genus *Empidonax*, whose distributions are known to include Chiapas. Although biological and geological evidence supports the idea of a barrier located at the Motagua-Polochic-Jocotán fault system in central Guatemala with phylogenetic implications for the biota (*Morrone, 2001*;

*Gutiérrez-García & Vázquez-Domínguez, 2013*; *Ríos-Muñoz, 2013*), our limited sampling cannot confirm the existence of a differentiated population in Mexico. We recommend further studies to assess population size and conservation status for this species in Mexico.

## ACKNOWLEDGEMENTS

We thank the institutions that provided information to the Global Biodiversity Information Facility. Thank you to the Secretaría de Medio Ambiente y Recursos Naturales for field work authorizations. We are grateful to Sean Rovito for excellent logistical skills that made our field expedition such a success. Thank you to Ana Isabel Bieler Antolín and the Microcine Lab, Facultad de Ciencias, UNAM for help with the photographs of specimens. Thank you to Michael L. P. Retter for providing feedback on our manuscript and directing us to information concerning the more recent sightings of *Elaenia frantzii* in Chiapas. Thank you to Willem-Pier Vellinga for directing us to the Xeno-canto recordings from Chiapas. We generated genetic data at the Evolutionary Genetics Laboratory, Museum of Vertebrate Zoology.

### Funding

The National Science Foundation funded field and laboratory work under grant no. DEB 1026393 to R.C.K.B. This material is based upon work supported by the National Science Foundation Graduate Research Fellowship under grant no. DGE 1106400 to Z.R.H. Any opinion, findings, and conclusions or recommendations expressed in this material are those of the authors and do not necessarily reflect the views of the National Science Foundation. Any use of trade, product, or firm names in this publication is for descriptive purposes only and does not imply endorsement by the U.S. Government. The funders had no role in study design, data collection and analysis, decision to publish, or preparation of the manuscript.

### Grant Disclosures

The following grant information was disclosed by the authors:
National Science Foundation: DEB 1026393.
National Science Foundation Graduate Research Fellowship: DGE 1106400.

### Competing Interests

The authors declare that they have no competing interests.

### Author Contributions

- Zachary R. Hanna conceived and designed the experiments, performed the experiments, analyzed the data, wrote the paper, prepared figures and/or tables, reviewed drafts of the paper.

- Marco F. Ortiz-Ramírez conceived and designed the experiments, performed the experiments, analyzed the data, wrote the paper, prepared figures and/or tables, reviewed drafts of the paper.
- César A. Ríos-Muñoz conceived and designed the experiments, performed the experiments, analyzed the data, wrote the paper, prepared figures and/or tables, reviewed drafts of the paper.
- Héctor Cayetano-Rosas conceived and designed the experiments, performed the experiments, analyzed the data, wrote the paper, prepared figures and/or tables, reviewed drafts of the paper.
- Rauri C. K. Bowie conceived and designed the experiments, contributed reagents/materials/analysis tools, reviewed drafts of the paper.
- Adolfo G. Navarro-Sigüenza conceived and designed the experiments, contributed reagents/materials/analysis tools, reviewed drafts of the paper.

### Animal Ethics

The following information was supplied relating to ethical approvals (i.e., approving body and any reference numbers):

UC Berkeley Institutional Animal Care and Use Committee approval number R317.

### Field Study Permissions

The following information was supplied relating to field study approvals (i.e., approving body and any reference numbers):

Secretaría de Medio Ambiente y Recursos Naturales permit number FAUT-0169.

### DNA Deposition

The following information was supplied regarding the deposition of DNA sequences:

GenBank accession numbers KU312259–KU312260.

### Data Deposition

Specimen CRH019 round skin and tissue sample deposited as MZFC:26000, Museo de Zoología "Alfonso L. Herrera," Facultad de Ciencias, Universidad Nacional Autónoma de México.

Specimen CRH019 tissue sample deposited as MVZ:Birds:188004, Museum of Vertebrate Zoology, University of California, Berkeley.

Specimen ZRH814 round skin and tissue sample deposited as MVZ:Birds:188003, Museum of Vertebrate Zoology, University of California, Berkeley.

Specimen ZRH814 tissue sample deposited as ZRH814, Museo de Zoología "Alfonso L. Herrera," Facultad de Ciencias, Universidad Nacional Autónoma de México.

### Supplemental Information

Supplemental information for this article can be found online at http://dx.doi.org/10.7717/peerj.1598#supplemental-information.

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
