# Peer review of "Phylogenetic and morphologic evidence confirm the presence of a new montane cloud forest associated bird species in Mexico, the Mountain Elaenia (Elaenia frantzii; Aves: Passeriformes: Tyrannidae)"

_PeerJ, doi:10.7717/peerj.1598_

## Round 0.1 · original submission · Minor Revisions

· Academic Editor

Minor Revisions

Dear authors

Thank you for submitting your manuscript to our journal. As you see our reviewers suggest a revision of your ms. If you are willing to do so, we would be happy to reconsider your revised manuscript.

Michael Wink

Reviewer 1 ·

Basic reporting

The paper entitled “Phylogenetic and morphologic evidence confirm the presence of a new montane cloud forest associated bird species in Mexico, the Mountain Elaenia (Elaenia frantzii; Aves: Passeriformes: Tyrannidae)” is interesting, straightforward, well written and deserves to be published in the PeerJ.

Experimental design

In particular, I value that the authors used different sources of evidence for their record.

Validity of the findings

The authors collected two birds and compared with other museum skins confirming morphologically the identity of the species. They also sequenced the mtDNA and show the position of the samples in the molecular phylogeny showing that they are grouped within the clade E. frantzii. Finally, they modelled the potential habitat that the species could occupy showing that the environment of the new record where the birds were collected is suitable for the presence of the species. So there is no doubt that the records are scientifically valid and belong to the species Elaenia frantzii.

Additional comments

One thing I think is needed is to add in the map the northernmost record of E. frantzii (the audio recording from northeastern, Utah, USA, deposited in VertNet), even if it is a non-breeding population; also, its biogeographical importance should be briefly discussed.

Line 39 ….”open humid”....
What do you mean by open humid forest? Please give more detailed characterization of this vegetation type.

Line 72
Please cite Figure 3

Line 87
Information about authorizations to collect and to do field work should go on the ACKS.

Line 90
Provide full name of the institutions, followed by the acronyms.

Line 160 ….
The record of E. frantzii in the USA should be added in the map and its biogeographical importance discussed.

Line 170
after “90.3913987°W, WGS84, Cobán, Departamento Alta Verapaz, Guatemala (Cahill, 2013)” cite Figure 3.

Reviewer 2 ·

Basic reporting

No Comments

Experimental design

No Comments

Validity of the findings

No Comments

Additional comments

The authors invested a lot of effort to prove that the range of Elaenia frantzii extends as far north as southern Mexico. The technical work has been performed according to standard and the paper is well written.

Minor issues:
33: New World flycatcher
44 Mexico,
52 tyrant flycatcher
64-72 remove parentheses from taxon authors as all taxa were described in Elaenia, add author for ultima
113 and more often: separate "°C" from preceding number by blank
130 see 44
140, 146, 148: use command citation() in R for correct referencing of the software (packages)
152 ten model replicates
155 ESRI
164 vs. 168: use consistent format for coordinates, I recommend "d° m' s" W"
188 cf. 113
205 ten replicates
fig2: do not italicize "New record"
fig3: italicize "Elaenia frantzii"
tab1: round values to 2 or 3 decimal places only

---

## Round 0.2 · accepted · Accept

· Academic Editor

Accept

Dear authors

Thank you for submitting your revision which covers the queries of the reviewers. Therefore, we accept your manuscript for publication. Thank you for submitting your ms to our journal.

Regards

Michael Wink
Editor